# OpenReview forum: "Chain-of-Agents: End-to-End Agent Foundation Models via Multi-Agent Distillation and Agentic RL"
_ICLR.cc/2026/Conference — Submitted to ICLR 2026_

### Official Review · Reviewer_fCmh · 2025-10-30

**Soundness:** 2
**Presentation:** 2
**Contribution:** 2
**Rating:** 4
**Confidence:** 3

**Summary:**

The paper proposes Chain-of-Agents (CoA), a framework for integrating multi-agent reasoning and tool use within a single model and execute end-to-end. Instead of running and orchestrating multiple interacting agents, CoA chain agents together and distills successful trajectories from expert multi-agent systems into a unified model.

The resulting system coordinates “role-playing agents” (planner, thinker, reflector, verifier) and “tool agents” (web search, wiki search, crawler, code executor), and finetune a base LLM using SFT/RL with synthesized agent trajectories.

Empirical results on a wide set of benchmarks (GAIA, BrowseComp, HLE, AIME, MATH500, etc.) show strong performance and efficiency gains compared to Tool-Integrated Reasoning (TIR) and multi-agent baselines.

**Strengths:**

- The paper presents a well-structured design combining multi-role reasoning and domain-specific tool agents under a single model.

- Extensive benchmarks across search, reasoning, math, and coding show consistent improvements.

- The paper provides substantial implementation details and appendices explaining the tool setup and data processing.

**Weaknesses:**

- The main technical contribution is the Chain-of-Agents architecture. The data synthesis pipeline and training (SFT + RL on distilled trajectories) mostly follows well-established practices.

- The paper demonstrates performance gains but offers limited insight into why it outperforms explicit multi-agent systems or where the improvement come from, as well as why sometimes the performance difference is uneven, why in some dataset improvement is significant while in some cases it is marginal. Or the gap between SFT and RL being different across benchmarks.

- The model is trained on Qwen 2.5, released in September 2024, even though Qwen 3 was public several months before submission (April 2025). This raises questions about the timeliness and potential performance ceiling.

- Some reported numbers are taken directly from prior work rather than rerun. For baselines that depend on web search results or use LLM-as-Judge scoring, direct reuse could compromise strict comparability as they can change over time (search results drift, judgment variance),

**Questions:**

- Some critical details (training data composition, tool configurations, trajectory sources) are relegated to the appendix; I feel it would be better to include these in main text directly.

- Why use Qwen 2.5 instead of Qwen 3, which was available before submission? Do you expect similar improvements if retrained on the newer backbone?

- For baselines whose results are borrowed rather than rerun, how do you control for changes in web search outcomes and stochastic LLM-as-Judge evaluations?

---

> ### Author Response · Authors · 2025-11-18
>
> We sincerely appreciate the reviewer fCmh for your careful evaluation of our paper and the constructive feedback. In response to your concerns, we have made several revisions to the manuscript. Please find the detailed changes below, and we hope they address the issues you raised. Based on Reviewer qAjm's suggestion, we have updated the model name from AFM to CoAM for clarity. However, to prevent any confusion, we have kept the original name AFM in our responses.
>
> ------
>
> **W1**:
>
> > The main technical contribution is the Chain-of-Agents architecture. The data synthesis pipeline and training (SFT + RL on distilled trajectories) mostly follows well-established practices.
>
> **Response:** We agree that our SFT and RL procedures follow established algorithms, and we do not claim novelty in the optimization methods themselves. Our contribution lies in the *Chain-of-Agents (CoA) reasoning paradigm* and the *agentic supervised fine-tuning (MAS→CoA distillation)* rather than modifying standard SFT/RL.
>
> **Contribution 1: Chain-of-Agents reasoning paradigm**
> The CoA framework restructures reasoning by assigning complementary role-playing agents that explore the problem from different perspectives while sharing a unified context. This enables coordinated planning, analysis, reflection, and verification, moving beyond the single-agent tool loop and forming the core of our contribution.
>
> **Contribution 2: Agentic supervised fine-tuning (MAS→CoA distillation)**
> We introduce a multi-agent distillation procedure that converts MAS trajectories into CoA-style traces and trains a single model to reproduce these structured behaviors end-to-end. This differs from TIR-style approaches (e.g., Search-R1 [1]), which rely on a single-agent iterative search loop rather than multi-role structured reasoning.
>
> **Additional design: Progressive quality filtering**
> Our progressive quality filtering is specifically designed for CoA trajectories. In particular, reflection enrichment and error-correction upsampling emphasize trajectories containing reflection and double-check behaviors, reinforcing these CoA roles during training.
>
> Overall, our main contribution lies in the CoA reasoning paradigm and the MAS→CoA distillation pipeline, whereas SFT/RL serve as established optimization tools rather than sources of novelty.
>
> ------
>
> **W2:**
>
> > The paper demonstrates performance gains but offers limited insight into why it outperforms explicit multi-agent systems or where the improvement come from, as well as why sometimes the performance difference is uneven, why in some dataset improvement is significant while in some cases it is marginal. Or the gap between SFT and RL being different across benchmarks.
>
> **Response:**
>
> **Why it outperforms explicit multi-agent systems and where the improvement comes from?**
> In the introduction, we address the limitations of multi-agent systems (MAS), including high costs, challenges in improving specific capabilities via data-driven training, and the fact that backbone models are not specifically trained for agentic tasks.
>
> Our model's performance advantage primarily stems from the following factors:
> 1. **Lower costs**: Each MAS agent requires independent context, whereas our CoA approach shares context across agents, reducing token consumption. As shown in Figure 3 of the paper, AFM reduces token usage by 84.6% and tool calls by 2/3 compared to a typical MAS.
>
> 2. **Data-driven training and agentic reasoning adaptation**: Table 5 of the revised paper shows AFM’s advantage over MAS in complex deep search tasks. The performance gain comes from distilling high-quality CoA trajectories, enabling the trained model to better adapt to the CoA reasoning paradigm and improve its role-playing and tool usage. Although the backbone of MASs, which refers to GPT-4.1 in the paper, may be stronger than our backbone models, it is not specifically trained for multi-agent tasks, which results in a less refined agent call ability based on tool returns.
>
> **Why sometimes the performance difference is uneven?**
>
> 1. **Dataset Step and Difficulty Variations**
> The performance gain varies across benchmarks due to differences in task difficulty and dataset distribution. For example, in **MHQA** and **Deep Search** tasks, as shown in Tables R1 and R2, the performance gains of AFM align closely with other methods. However, tasks like **BrowseComp** and **MusiQue** show smaller gains, likely due to their inherent complexity. In contrast, simpler tasks like **GAIA** and **WebWalker** show larger improvements, as AFM's tool usage is more effective in these cases.
>
> This discrepancy can be attributed to task complexity: in simpler tasks, AFM significantly benefits from optimized tool usage, while in more complex tasks, all models show similar performance due to the greater challenge posed by the task.

---

> > ### Author Response · Authors · 2025-11-18
> >
> > 2. **SFT vs RL Performance Gain Difference**
> > We compare the performance gains of **AFM** and other models in the **SFT** and **RL** stages, as shown in Tables R3 and R4 (i.e., SFT relative to the baseline and RL relative to SFT). For tasks like Math and Deep Search, the performance gains in the SFT stage are similar between AFM and other models, and the same holds true for the RL stage. To summarize, we think the intrinsic data distribution make SFT and RL exhibit significant gain difference across various benchmarks.
> >
> >
> > Table R1: Performance Gain Comparison on Deep Search Tasks (Qwen2.5-7B-instruct based models)
> > | Method \ Benchmarks | GAIA | WebWalker | BrowseComp |
> > |---------------------|------|-----------|------------|
> > | WebShaper [2]       | 45.6 | 48.3      | -          |
> > | WebSailor [3]       | 46.3 | -         | 9.9        |
> > | AFM                 | 53.3 | 59.9      | 14.9       |
> >
> > Table R2: Performance Gain Comparison on MHQA Tasks (Qwen2.5-7B-instruct based models)
> > | Method \ Benchmarks | NQ   | TriviaQA | PopQA | HotPotQA | 2Wiki | MusiQue | Bamboogle |
> > |---------------------|------|----------|-------|----------|-------|---------|-----------|
> > | Search-R1 [1]       | 27.7 | 25.2     | 38.5  | 20.6     | 19.2  | 9.8     | 22.4      |
> > | AFM                 | 32.3 | 27.7     | 45.3  | 27.5     | 27    | 17.5    | 35.2      |
> >
> > Table R3: Performance Gain Comparison on Math Benchmarks
> > | Method \ Benchmarks | AIME24 | AIME25 |
> > |---------------------|--------|--------|
> > | AFM-SFT             | 27.7   | 27.9   |
> > | Retool-SFT [4]      | 26.3   | 19.4   |
> > | AFM-RL              | 25.7   | 28.8   |
> > | Retool-RL [4]       | 26.1   | 14.8   |
> >
> > Table R4: Performance Gain Comparison on Deep Search Benchmarks
> > | Method \ Benchmarks | BrowseComp | GAIA  | WebWalker |
> > |---------------------|------------|-------|-----------|
> > | AFM-SFT             | 13.8       | 50.5  | 58.4      |
> > | Websailor-SFT [3]   | 6.6        | 33.0  | -         |
> > | WebThinker-SFT [5]  | -          | 37.9  | 38.8      |
> > | AFM-RL              | 1.1        | 2.9   | 1.5       |
> > | Websailor-RL [3]    | 3.3        | 6.6   | -         |
> > | WebThinker-RL [5]   | -          | 3.8   | 4.6       |
> >
> > ------
> >
> > **W3 and Q2:**
> >
> > Weakness 3:
> > > The model is trained on Qwen 2.5, released in September 2024, even though Qwen 3 was public several months before submission (April 2025). This raises questions about the timeliness and potential performance ceiling. Why use Qwen 2.5 instead of Qwen 3, which was available before submission? Do you expect similar improvements if retained on the newer backbone?
> >
> > Question 2
> > > Why use Qwen 2.5 instead of Qwen 3, which was available before submission? Do you expect similar improvements if retrained on the newer backbone?
> >
> > **Response:** We selected Qwen 2.5 as the baseline model primarily to ensure fair comparison, as most concurrent works in this paper also use models from the Qwen 2.5 family, expecially Qwen-2.5-32B-Instruct, and QwQ-32B. This helps eliminate bias from differences in the backbone models.
> >
> > We acknowledge your concerns regarding the timeliness and performance bottlenecks of the backbone. To address these issues, we trained Qwen 3 models using the training dataset presented in this paper and evaluated their performance on the most challenging benchmark, BrowseComp. As shown in Table R3, Qwen 3 significantly outperforms Qwen 2.5 models with similar backbone parameter sizes—this verifies that our method can still yield performance gains on difficult search tasks. Additionally, we conducted supplementary experiments by training the Qwen-2.5-72B-Instruct model on the same training dataset. The results, reported in Table R3, demonstrate the model’s potential for scaling.
> >
> > Table R3: Performance Comparison of Qwen3 and Qwen2 Models Before and After SFT with CoA data
> > | Model                  | BrowseComp |
> > |------------------------|------------|
> > | Qwen2.5-32B-SFT        | 14.4       |
> > | Qwen2.5-72B-SFT        | 30.9       |
> > | Qwen3-32B-SFT          | 15.5       |
> > | Qwen3-30B-A3B-SFT      | 17.1       |

---

> > > ### Comment · Reviewer_fCmh · 2025-11-28
> > >
> > > Thanks for the additional result, but seem the number for after SFT with CoA data is missing?

---

> ### Author Response · Authors · 2025-11-18
>
> **W4 and Q3:**
>
> Weakness 3:
> > Some reported numbers are taken directly from prior work rather than rerun. For baselines that depend on web search results or use LLM-as-Judge scoring, direct reuse could compromise strict comparability as they can change over time (search results drift, judgment variance),
>
> Question 3:
> > For baselines whose results are borrowed rather than rerun, how do you control for changes in web search outcomes and stochastic LLM-as-Judge evaluations?
>
> **Response:**
> As you pointed out, the baselines we compare with in the deep search tasks use web search and LLM-as-Judge evaluations. However, we would like to clarify that in our study, the **LLM-as-Judge** uses the open-source Qwen-2.5-72B-Instruct model as the evaluator. This setup ensures that:
> 1. Model parameters do not change over time, preventing evaluation results from drifting.
> 2. Both **Webshaper [2]** and **WebSailor [3]** use the same setup, ensuring fairness in comparison.
>
> We acknowledge that web search APIs may be subject to time-related shifts in outcomes. However, to clarify, we re-ran the open-source models on the deep search tasks during our experiment. The results in Table R4 show that the reproduced results were generally lower than those reported in the original papers. Therefore, in order to compare with the best performance, we only included the missing benchmark results in Table 5 of our revised manuscript (marked in gray). Thus, we believe our comparison ensures fairness to the best extent possible.
>
> Table R4: Performance on Deep Search tasks of open-source models or open-source multi-agent systems. (*) indicates reproduced results, while non-asterisk entries are reported in the original papers.
> | Method                        | GAIA  | BrowseComp | HLE  | WebWalker |
> |-------------------------------|-------|------------|------|-----------|
> | OWL (*)                        | 53.6  | -          | 6.4  | 10.2      |
> | OAgents (*)                    | 58.3  | 13.7       | 20.2 | -         |
> | search-o1 Qwen2.5-32B-Instr (*)| 28.2  | -          | -    | -         |
> | search-o1 QwQ-32B (*)          | 39.8  | -          | 10.8 | 34.1      |
> | webthinker QwQ-32B             | 48.5  | -          | 15.8 | -         |
> | webthinker QwQ-32B (*)         | 39.8  | 2.8        | 10.2 | 46.5      |
> | webdancer QwQ-32B             | 51.5  | 3.8        | -    | 47.9      |
> | webdancer QwQ-32B (*)         | 42.7  | 3.1        | 7.2  | 45.4      |
>
> ------
>
> **Q1:**
> > Some critical details (training data composition, tool configurations, trajectory sources) are relegated to the appendix; I feel it would be better to include these in main text directly.
>
> **Response:**
> Due to space constraints, the data sources, tool configurations, and other details you mention were placed in the appendix. The main reasons are:
> 1. Our work involves data from dozens of sources across multiple domains, and the tools used are diverse, which would occupy considerable space if included in the main text.
> 2. We believe that presenting the methodology and experimental discussion in the main text more effectively conveys our motivation and contributions.
>
> Fortunately, the camera-ready version allows an additional page. We have compressed and extracted some of the most critical information from the appendix and added it to Section 4.1 in the revised manuscript. Please see the revised version in the attachment.
>
>
> ------
>
> **Reference**:
>
> [1] Jin, B., Zeng, H., Yue, Z., Yoon, J., Arik, S., Wang, D., ... & Han, J. (2025). Search-r1: Training llms to reason and leverage search engines with reinforcement learning. arXiv preprint arXiv:2503.09516.
>
> [2] Tao, Z., Wu, J., Yin, W., Zhang, J., Li, B., Shen, H., ... & Zhou, J. (2025). Webshaper: Agentically data synthesizing via information-seeking formalization. arXiv preprint arXiv:2507.15061.
>
> [3] Li, K., Zhang, Z., Yin, H., Zhang, L., Ou, L., Wu, J., ... & Zhou, J. (2025). WebSailor: Navigating Super-human Reasoning for Web Agent. arXiv preprint arXiv:2507.02592.
>
> [4] Feng, J., Huang, S., Qu, X., Zhang, G., Qin, Y., Zhong, B., ... & Zhong, W. (2025). Retool: Reinforcement learning for strategic tool use in llms. arXiv preprint arXiv:2504.11536.
>
> [5] Li, X., Jin, J., Dong, G., Qian, H., Wu, Y., Wen, J. R., ... & Dou, Z. (2025). Webthinker: Empowering large reasoning models with deep research capability. arXiv preprint arXiv:2504.21776.
>
> -----
>
> We would appreciate it if you could let us know whether your concerns are addressed by our response. We hope the revised manuscript will lead you to reconsider your evaluation and adjust your score accordingly.

---

> ### Author Response · Authors · 2025-11-28
>
> Thank you for your follow-up question and for giving us the opportunity to clarify the results more precisely. We sincerely apologize for the confusion caused by the mislabeled title in Table R3.
>
> **Confirmation of the reported performance**
> All results presented in Table R3 are **after SFT with CoA data**. The previous title incorrectly implied a comparison between before-SFT and after-SFT results, which was not the case. We have corrected the table title in the revised response to ensure the meaning is unambiguous.
>
> **Why baseline (before-SFT) numbers are not listed**
> We did not include the raw (pre-SFT) performance of Qwen3 because previous studies have shown that Qwen3 models underperform on challenging deep-search benchmarks. For instance, prior work [6] reports that Qwen3-32B achieves a score of only 12 on xBench-DeepSearch. Additionally, Qwen3-235B-A22B-2507 scores just 8, 44.7, and 45.5 on BrowseComp, GAIA, and xBench-DS respectively [7]. These results suggest that its out-of-the-box performance on complex search tasks is limited. Including such low baseline scores would not offer new insights beyond what is already documented in the existing literature.
>
>
> **Additional benchmarks for completeness**
> To strengthen our response, we extended Table R3 with results from two more challenging benchmarks—**GAIA** and **xBench-DS**—in addition to **BrowseComp**. These results consistently show that CoA data improves both Qwen2.5 and Qwen3 models.
>
> Updated Table R3 is shown below:
>
> **Revised Table R3. Performance of Qwen2.5 and Qwen3 Models After SFT with CoA Data**
>
> | Model              | Stage      | BrowseComp | GAIA | xBench-DS |
> |-------------------|------------|------------|------|-----------|
> | Qwen2.5-32B-Ins    | After SFT | 14.4       | 57.3 | 47.0      |
> | Qwen2.5-72B-Ins    | After SFT | 30.9       | 62.1 | 54.0      |
> | Qwen3-32B          | After SFT | 15.5       | 64.1 | 51.0      |
> | Qwen3-30B-A3B      | After SFT | 17.1       | 59.2 | 49.0      |
>
> We have included the experimental results and analysis in Appendix B.3.7. We hope this clarification resolves the issue, and we thank you again for your careful reading and insightful questions.
>
> **Reference**
>
> [6] Yu, C., Lu, S., Zhuang, C., Wang, D., Wu, Q., Li, Z., ... & Lin, T. (2025). *Aworld: Orchestrating the training recipe for agentic AI.* arXiv:2508.20404.
>
> [7] Zeng, W., He, K., Kuang, C., Li, X., & He, J. (2025). Pushing test-time scaling limits of deep search with asymmetric verification. arXiv preprint arXiv:2510.06135.

---

### Official Review · Reviewer_AH31 · 2025-10-31

**Soundness:** 3
**Presentation:** 3
**Contribution:** 3
**Rating:** 6
**Confidence:** 3

**Summary:**

The paper presents a method for training a single LLM that integrates tool usage, enabled by data generated by multiple agents. The key idea is to create training data using a multi-agent setup, where each agent has access to specific tools. The interactions of these agents can be represented in the training data through special tokens. This data is then used to perform supervised fine-tuning or RL-based post-training on a single distilled LLM, with an LLM-as-a-judge component providing the reward signal for RL training. The effectiveness of the approach is evaluated across multiple backbones of varying sizes.

**Strengths:**

1. The main strength of this paper lies in its experimental section. The results are compelling, with strong evaluations against numerous baselines and models at different scales, including 32B ones. These elements collectively support the paper’s claims effectively.
2. The analysis in Section 5 highlights important contributing factors. It is particularly interesting to observe that test-time computation significantly benefits the trained network, although further clarifications would be helpful (see questions below).
3. The manuscript is well organized and clearly written.

**Weaknesses:**

1. The novelty of the work does not appear to be particularly strong; it seems more like a well-executed engineering contribution. My main concern arises from the comparison with Search-R1. The proposed framework resembles an N-agent extension of Search-R1, where multiple agents with different tools replace the single agent equipped with a search tool. While effective in practice, to me this looks to be quite incremental.
2. Another limitation is that distilling all capabilities into a single LLM restricts the integration of new tools, a process that is much more straightforward in traditional roleplay-based agentic setups. Is there any possible way to mitigate this limitation?

**Questions:**

1. Is there any intuition on why test-time adaptation performs better with the proposed approach than with the baselines as claimed?
2. Can the authors clarify where the novelty stands relative to Search-R1?
3. Could the system be made more flexible to accommodate new tools or agent roles? This does not require experiments, but rather a discussion.

---

> ### Author Response · Authors · 2025-11-18
>
> Our sincere gratitude goes to reviewer AH31 for the detailed and valuable comments provided. You have significantly contributed to enhancing the clarity and depth of our work. We have incorporated your suggestions in the revised version of our manuscript. In response to Reviewer qAjm's suggestion, we have renamed our model from AFM to CoAM for clarity. However, to avoid any confusion, we have retained the original name AFM in response.
>
> ------
>
> **W1 and Q2:**
>
> Weakness 1:
> > The novelty of the work does not appear to be particularly strong; it seems more like a well-executed engineering contribution. My main concern arises from the comparison with Search-R1. The proposed framework resembles an N-agent extension of Search-R1, where multiple agents with different tools replace the single agent equipped with a search tool. While effective in practice, to me this looks to be quite incremental.
>
> Question 2:
> > Can the authors clarify where the novelty stands relative to Search-R1?
> **Response:**
> While it may appear at first glance that our framework is an N-agent extension of Search-R1, we would like to clarify the following points:
>
> 1. **Conceptual and methodological differences**: Search-R1 primarily emphasizes the TIR paradigm, in which incorporating additional callable tools is indeed an incremental improvement. The differences of our approach are as follows:
> - Our proposed **Chain of Agents (CoA)** paradigm further emphasizes the importance of role-playing agents. Different roles explore the answer from diverse perspectives during reasoning while sharing context, enabling planning, thinking, reflection, and verification processes similar to multi agent system (MAS) behavior.
>  - We introduce a **multi-agent distillation** method, which first converts MAS trajectories into CoA style trajectories and then distills them into a single model, allowing the model to exhibit end-to-end MAS-like problem-solving capabilities. This approach is fundamentally different from the intent of Search-R1 and similar works.
>
> 2. **Practical benefits of our method**: We extracted a subset of code and math questions (6,000) from our training data and constructed training sets using both the TIR approach and our CoA approach. We then performed SFT on the Qwen-2.5-7B-Instruct model and evaluated its performance on several code and math benchmarks. The results are presented in Tables R1.
> Additionally, the training data sources for MHQA tasks are the same as those used in Search-R1. Therefore, we directly extracted the corresponding results from Table 7 of the revised manuscript and included them in Table R2 to compare the performance of our method with that of Search-R1 on the MHQA tasks.
> The results in Tables R1 and R2 demonstrate that the SFT-trained model consistently outperforms the TIR-trained model, highlighting the effectiveness of generating data through MAS to construct CoA trajectories and applying multi-agent system distillation.
>
> Table R1: Comparison on mathematical and code tasks. Pass@1 is reported; mean@16 for AIME24/25. Evaluation settings follow those described in Subsection B.2 of the paper.
> | Method / Benchmarks      | AIME24 | AIME25 | MATH500 | LCB v4 | LCB v5 |
> |---------------------------|--------|--------|---------|--------|--------|
> | Qwen-2.5-7B-Instruct      | 7.5    | 2.92   | 63.6    | 15.84  | 12.57  |
> | TIR SFT                   | 6.89   | 6.89   | 63.2    | 14.85  | 17.37  |
> | CoA SFT                   | 18.54  | 15.42  | 64.4    | 18.81  | 17.96  |
>
> Table R2: Comparison on MHQA tasks. Pass@1 is reported. Evaluation settings follow those described in Subsection B.2 of the paper.
> | Method / Benchmarks        | NQ   | TrivaQA | PopQA | HotPotQA | 2Wiki | MusiQue | Bamboogle |
> |-----------------------------|------|---------|-------|----------|-------|---------|-----------|
> | Qwen-2.5-7B-Instruct        | 11.6 | 35.6    | 1.2   | 16.4     | 22.2  | 4.8     | 14.4      |
> | TIR SFT[^1]                 | 31.8 | 35.4    | 12.1  | 21.7     | 25.9  | 6.6     | 11.2      |
> | CoA SFT                     | 39.8 | 59.6    | 39.3  | 38.8     | 50.7  | 19.5    | 44.4      |
>
> Furthermore, we have included the corresponding experiments in Appendix B3.2 of the revised manuscript for additional details.

---

> > ### Author Response · Authors · 2025-11-18
> >
> > ------
> >
> > **W2 and Q3:**
> >
> > W2:
> > > Another limitation is that distilling all capabilities into a single LLM restricts the integration of new tools, a process that is much more straightforward in traditional roleplay-based agentic setups. Is there any possible way to mitigate this limitation?
> >
> > Q3:
> > > Could the system be made more flexible to accommodate new tools or agent roles? This does not require experiments, but rather a discussion. Is there any possible way to mitigate this limitation?
> >
> > **Response:**
> > Our SFT and RL training do not compromise the model’s underlying instruction-following abilities. By adding descriptions of new tools and roles in the prompt, the model can still adapt and exhibit the corresponding role-playing and tool-invocation capabilities.
> >
> > Although additional experiments were not required, we conducted a simple prompt-only evaluation to demonstrate this flexibility. Specifically, we tested the AFM-32B-RL trained with math and code data by introducing a **new multimodal OCR tool** (`visual_inspector`) and two additional roles (`double_check` and `suggested_answer`) that were not seen during training. The evaluation set consists of 26 GAIA questions with images[^2]. As a baseline, we used the Qwen-32B-Instruct model with the same prompts. The evaluation metric is **Pass@1, mean@3**. The prompt and tool specifications are provided below （also provided in Appendix D.6 of the revised manuscript）.
> >
> > Table R3: Performance comparison between Qwen-2.5-32B-Instruct and AFM-32B-RL on 26 multi-modal GAIA questions (Accuracy of three independent experiments and their average).
> >
> > | Method / Acc                | 1st exp  | 2nd exp  | 3rd exp   | Avg.    |
> > |-----------------------------|-------|-------|--------|--------|
> > | Qwen-2.5-32B-Instruct       | 11.53 | 11.53 | 15.38  | 12.82  |
> > | AFM-32B-RL                  | 19.23 | 19.23 | 23.08  | 20.51  |
> >
> > Results in Table R3 indicate that AFM-32B-RL outperforms the baseline, demonstrating that even though the training data did not include these new tools and roles, the AFM-32B-RL model can better utilize them, rather than being restricted in its use of tools or roles.
> >
> > This experiment and analysis is also added in Appendix B.3.6 of the revised manuscript.
> >
> > We extended the prompts from Appendix D.2 to include the new tools and roles below:
> > > 7. **suggested_answer**: Based on the historical trajectory, you can come up with a suggested answer without checking the answer again.
> > > 8. **double_check**: After giving the suggested answer, reflect on your historical trajectory and provide reasoning based on the credibility of the suggested answer. If you are not confident, rethink and replan to solve the task; otherwise, provide your answer.
> > > 9. **visual_inspector**: When analyzing or understanding image content, use this function. The image will be processed by an external multimodal LLM and returned as text. Format:
> > > ```json
> > > <visual_inspector>
> > > {"file_path": "File path or web image URL (e.g., 'https://example.com/image.jpg') to be read as an image. Must be in supported formats (.jpg/.jpeg/.png/.gif/.bmp/.webp).", "question": "[Optional] Your question about the image content. Provide as much context as possible. Do not pass this parameter if you just want a description of the image."}
> > > </visual_inspector>
> > > ```
> >
> > **Mitigating this limitation**
> > Although AFM shows better adaptation to new tools and roles, absolute performance on this evaluation remains lower than on tasks with fully trained tools and roles (e.g., code tools + multiple roles). To achieve comparable performance, these new tools and roles would need to be included in a new AFM training cycle, involving trajectory collection and multi-agent distillation.
> >
> >
> > ------
> >
> > **Q1:**
> > > Is there any intuition on why test-time adaptation performs better with the proposed approach than with the baselines as claimed?
> >
> > **Response:**
> > Intuitively, the presence of multiple role-playing agents in the CoA paradigm encourages exploration of diverse problem-solving strategies compared to the simpler TIR framework. In particular, the **reflection and double_check** allows different trajectories to reconsider and potentially revise previous decisions, which can lead to significantly different behaviors across trajectories.
> >
> > As a result, under the same sampling temperature, CoA is more likely to generate **diverse trajectories** than TIR. This increased diversity enhances the probability that **best@N sampling** contains a correct answer, providing a practical explanation for the observed improvements in test-time adaptation.

---

> > > ### Author Response · Authors · 2025-11-18
> > >
> > > **Footnote:**
> > >
> > > [^1] The result is obtained from Search-r1 [1], as we use same training data sources.
> > > [^2] https://huggingface.co/datasets/gaia-benchmark/GAIA
> > >
> > > **Reference:**
> > >
> > > [1] Jin, B., Zeng, H., Yue, Z., Yoon, J., Arik, S., Wang, D., ... & Han, J. (2025). Search-r1: Training llms to reason and leverage search engines with reinforcement learning. arXiv preprint arXiv:2503.09516.
> > >
> > > ------
> > >
> > > We hope the response and the updated manuscript meets your expectations. We hope you might consider increasing your score based on these improvements.

---

> ### Comment · Area_Chair_Vf1N · 2025-11-28
> **Reminder: Engage with Authors During Rebuttal**
>
> Quick reminder: the rebuttal period is still open, and the deadline is in less than one week. Please continue the discussion with the authors and share any clarifications or updates to your assessment before the rebuttal closes.

---

### Official Review · Reviewer_qAjm · 2025-11-02

**Soundness:** 3
**Presentation:** 3
**Contribution:** 3
**Rating:** 4
**Confidence:** 3

**Summary:**

This paper proposes an agentic pipeline in which a single model can act as multiple agents to simulate a "multi-agent" workflow. The proposed procedure involves distilling multi-agent traces into a single model, which the authors refer to as an "agent foundation model" (AFM). The proposed chain of agents approach achieves state of the art performance when compared to single-agent tool integrated reasoning (TIR) pipelines.

**Strengths:**

- The empirical results of this work are strong, with state-of-the-art performance across a variety of benchmarks across various applications, including deep search, math, and MHQA tasks.
- The token reduction compared to multi-agent systems is significant, reducing inference cost (as measured in tokens) by an impressive 84%.
- The ablations suggest that both the SFT and RL components of the training pipeline contribute meaningfully, which is a useful ablation for understanding which components of the proposed technique matter.

**Weaknesses:**

- The framing around AFMs seems to be undermined, to some degree, by the fact that the proposed AFMs are domain-specific, requiring specialized AFMs (specialization is at odds with the colloquial meaning of the term "foundation model"). Section 5.3 described one example in which cross-domain generalization was observed, but this does not seem like strong enough evidence to claim that the proposed models are general-purpose agentic foundation models.
- I may have missed something, but it remains unclear to me whether performance gains stem from the chain of agents technique itself or primarily from higher-quality training data obtained via multi-agent distillation and filtering. A prompting-only baseline applying chain of agents format to existing models would help to disentangle these potential contributions.
- The method relies heavily on LLM as a judge, which might raise questions about the scalability of the proposed method -- is the performance ceiling capped by the best available frontier model, for a given type of task?

**Questions:**

- The term "agent foundation model" is perhaps overstated, as the proposed agents are still task-specific. Can the authors elaborate on this terminology choice? How are agent foundation models different from foundation models that process text, more generically?
- Note that most of the citations in the paper are in-text, but many of them should actually be parenthetical.

---

> ### Author Response · Authors · 2025-11-18
>
> We would like to express our heartfelt thanks to reviewer qAjm for the thoughtful and constructive feedback. Your comments have been instrumental in refining our manuscript. We have carefully considered all your suggestions and have submitted a revised version.
>
> ------
>
> **Q1 and W1:**
>
> Weakness 1:
> > The term "agent foundation model" is perhaps overstated, as the proposed agents are still task-specific. Can the authors elaborate on this terminology choice? How are agent foundation models different from foundation models that process text, more generically?
>
> Question 1:
>
> > The term "agent foundation model" is perhaps overstated, as the proposed agents are still task-specific. Can the authors elaborate on this terminology choice? How are agent foundation models different from foundation models that process text, more generically?
>
> **Response:** We acknowledge that our present models remain domain-specific due to their reliance on domain-tailored tools and training data. We agree that this limits their ability to be considered “foundation-level” in the same sense as large text-based foundation models.
>
> **What is agent foundation model (AFM)?**
> Our envisioned AFM differs from generic text foundation models in that it aims to support end-to-end, role-adaptive, multi-tool agentic behavior, resembling a lightweight multi-agent system within a single model.
>
>
> **Why we initially used the AFM terminology?**
> The models presented in this paper are early prototypes of this broader vision. They exhibit role playing and dynamic tool usage within the chain-of-agents (CoA) reasoning framework, albeit within constrained domains. We adopted the AFM term to emphasize two distinctions from traditional tool integrated reasoning (TIR) models:
> - CoA-based agentic reasoning: Unlike TIR models that operate with a single tool interface and a single reasoning role, our models learn to coordinate multiple roles and tools throughout the reasoning process.
> - Coverage across multiple domains: We evaluated models across three diverse domains (Deep Search, Math & Code, and MHQA). While each AFM is specialized, the overall method demonstrates that our metohds can be effective beyond a single task family.
>
> **Manuscript Revision:**
> We agree that the term "agent foundation model" may overstate the generality of the current system. To more accurately reflect its capabilities, we have updated the terminology in the revised manuscript. The term "Chain-of-Agents Model  (CoAM)" now better captures the core features without implying a foundational scope.
>
>
> ------
>
> **Q2:**
> > Note that most of the citations in the paper are in-text, but many of them should actually be parenthetical. \citet改\citep
>
> **Response:**  We have revised the manuscript according to your suggestion and updated the citations to use parenthetical style where appropriate.
>
> ------
>
> **W2:**
>
> > I may have missed something, but it remains unclear to me whether performance gains stem from the chain of agents technique itself or primarily from higher-quality training data obtained via multi-agent distillation and filtering. A prompting-only baseline applying chain of agents format to existing models would help to disentangle these potential contributions.
>
> **Response:** Both the chain of agents technique and multi-agent distillation contribute to improvements, but for the trained models in our paper, the primary gains indeed come from multi-agent distillation.
>
> **Prompt-only experiments to illustrate the source of performance gain**
>
> To directly address the reviewer’s concern, we added prompt-only baselines applying the CoA prompt to an untrained backbone model (Qwen-2.5-Coder-7B). We additionally provide a TIR-style prompt-only baseline and include the results of the trained model after distillation. The TIR prompt is similart to that in Retool [1]. Table R1 summarizes results across code and math benchmarks. This experiment has also been added to Appendix B.3.1 of the revised manuscript.
>
> Table R1: Comparison of prompt-based baselines, backbone model, and CoA-SFT model on math/code benchmarks. Pass@1 is reported; mean@16 for AIME24/25 and AMC23. Evaluation settings follow those described in Subsection B.2 of the paper.
> | Method     | AIME24 | AIME25 | MATH500 | AMC23 | OlympiadBench |  LCB v4| LCB v5| CodeContests |
> |------------|--------|--------|---------|-------|---------------|---|---|---|
> | Backbone |  7.5   | 2.92   |  68.6   |   16.4    |  11.9    |  15.84 | 17.96  |   0.0     |
> | TIR Prompt |  9.17  |  4.58  |  40.6   |   37.73    |   30.22   |  11.88  | 9.58  |    4.24  |
> | CoA Prompt |  11.46 | 10.00  |  53.8   |    48.36   |   35.70   |  12.87  | 14.37  |    5.46   |
> | After distillation |   27.5 |15.4 |74.0| 60.3 |40.3     | 28.0  | 26.3  | 13.3  |

---

> > ### Author Response · Authors · 2025-11-18
> >
> > **Findings**
> > 1. ***CoA prompt-only yields mixed and unstable improvements***
> > - On some tasks, mostly difficult tasks, the backbone model lacks baseline capabilities, and CoA prompts + tools can provide additional information, leading to gains.
> > - On moderate tasks, the untrained CoA format can introduce confusion (e.g., incorrect tool calls), and the tool usage is not strong enough to compensate. Therefore, the performance of CoA Prompt-only method may drop relative to the base model.
> > 2. ***CoA prompt is generally stronger than TIR prompt***
> >  Despite both suffering from untrained tool-use patterns, CoA’s role-playing agents tends to activate more robust problem-solving strategies. This sometimes mitigates tool-use errors and leads to modest but consistent gains over TIR prompting.
> > 3. ***multi-agent distillation provides the dominant performance improvements***
> >  The model after distillation shows large and stable gains across all benchmarks. This confirms that:
> > - The CoA paradigm itself is helpful but limited without training.
> > - Teaching the model to think in a CoA paradigm through multi-agent distillation is essential for strong performance.
> >
> >
> > ------
> >
> > **W3:**
> > > The method relies heavily on LLM as a judge, which might raise questions about the scalability of the proposed method -- is the performance ceiling capped by the best available frontier model, for a given type of task?
> >
> > **Response:**
> > While our method employs LLM-as-judge, the evaluation task is relatively simple: the judge only assesses short strings enclosed within `<answer>` tags in generated trajectories. This is effectively a string-level correctness check rather than complex reasoning. As a result, the performance ceiling of our models is not capped by the judge’s capability. Even less advanced LLMs can reliably perform this evaluation, and using a more capable LLM does not significantly change the assessment. Thus, the evaluation remains robust and scalable.
> >
> > Additionally, we would like to clarify why and for which tasks we employ LLM-as-judge:
> > **LLM as a judge is applied only to Deep Search tasks**
> > For other benchmarks, we use alternative methods: Exact Match (EM) for MHQA and the MathVerify[^1] library for code and mathematical reasoning. This design ensures both accurate evaluation and fair comparison with prior works.
> >
> > **Reason for applying LLM as a judge**
> > - Accurate assessment: For simple answer formats (names, multiple-choice, integers), EM suffices. However, for deep research tasks (e.g., GAIA, BrowseComp), EM can be biased: semantically equivalent answers may be penalized for minor lexical differences (e.g., 3.14 vs 3.14159 for π). LLM as a judge evaluates such answers more holistically, mitigating this bias.
> > - Fair comparison: Our configuration, including choice of LLM, prompts (Subsection D.4 in the manuscript), strictly follows prior works such as WebShaper [2] and WebSailor [3]. We use the open-source Qwen-2.5-72B-instruct as the judge LLM, which is stable and deterministic, ensuring reproducible results independent of model frontier changes.
> >
> > ------
> >
> > **References**
> >
> > [1] Feng, J., Huang, S., Qu, X., Zhang, G., Qin, Y., Zhong, B., ... & Zhong, W. (2025). Retool: Reinforcement learning for strategic tool use in llms. arXiv preprint arXiv:2504.11536.
> >
> > [2] Tao, Z., Wu, J., Yin, W., Zhang, J., Li, B., Shen, H., ... & Zhou, J. (2025). Webshaper: Agentically data synthesizing via information-seeking formalization. arXiv preprint arXiv:2507.15061.
> >
> > [3] Li, K., Zhang, Z., Yin, H., Zhang, L., Ou, L., Wu, J., ... & Zhou, J. (2025). WebSailor: Navigating Super-human Reasoning for Web Agent. arXiv preprint arXiv:2507.02592.
> >
> > **Footnote**
> >
> > [^1] https://github.com/huggingface/Math-Verify
> >
> > ------
> >
> > We would appreciate it if you could let us know if our response addresses your concerns. We hope that you might consider raising your score in light of these clarifications.

---

> ### Comment · Area_Chair_Vf1N · 2025-11-28
> **Reminder: Engage with Authors During Rebuttal**
>
> Quick reminder: the rebuttal period is still open, and the deadline is in less than one week. Please continue the discussion with the authors and share any clarifications or updates to your assessment before the rebuttal closes.

---

### Author Response · Authors · 2025-12-02

Dear reviewers and ACs,

Thank you for your constructive feedback. We appreciate that reviewers consistently highlighted several core strengths of the paper:

1. Strong empirical results with state-of-the-art performance across various benchmarks and applications (Reviewer qAjm, Reviewer AH31, Reviewer fCmh).
2. Meaningful contributions from both the SFT and RL components, clearly identified through ablation studies (Reviewer qAjm).
3. Comprehensive experimental evaluations against numerous baselines, including large-scale models (Reviewer AH31, Reviewer fCmh).
4. Well-organized manuscript and substantial implementation details (Reviewer AH31, Reviewer fCmh).
5. Significant token reduction compared to multi-agent systems, reducing inference costs by 84% (Reviewer qAjm).

In response to the reviewers' comments, we have made several important revisions and additions:

1. **Terminology Clarification**: We updated the term "Agent Foundation Model" (AFM) to "Chain-of-Agents Model" (CoAM) to better reflect the domain-specific nature of our models, as reflected in the revised manuscript.
2. **Performance Gain Attribution**: We conducted prompt-only and SFT experiments comparing CoA with TIR to clarify the contributions of the CoA technique versus multi-agent distillation. Additionally, we further emphasized that the core contribution lies in the CoA reasoning paradigm and the MAS distillation approach, as detailed in Appendices B.3.1 and B.3.2.
3. **LLM as a Judge & Fairness Concerns**: We explained that the LLM is a fixed-parameter open-source model and that the performance of other open-source models has been re-evaluated to ensure fairness and comparability.
4. **Flexibility with New Tools and Roles**: We demonstrated the model's flexibility in adapting to new tools and roles through prompt extensions. This was illustrated by experiments with a new multimodal OCR tool and additional roles, as shown in Appendix B.3.6.
5. **Adaptation to SOTA Open-Source Models**: We added new experiments comparing the performance of models trained with CoA data on Qwen 2.5 and Qwen 3, particularly on BrowseComp, to verify that our method can still yield performance gains on challenging tasks (see Appendix B.3.7).

We sincerely appreciate Reviewer fCmh for their thoughtful feedback, which helped us refine the manuscript. We have addressed their concerns by providing additional clarifications and performance comparisons across different benchmarks. Unfortunately, due to system-related issues, we did not receive follow-up feedback from reviewers AH31 and qAjm. Nonetheless, we have proactively addressed all their concerns in the revised manuscript.

We once again thank all reviewers, ACs, and PCs for their valuable time, feedback, and participation throughout the process.

---

### Meta-Review · Area_Chair_aiud · 2026-01-07

**Summary:**

The paper proposes "Chain-of-Agents" (CoA), a framework that distills the reasoning patterns of multi-agent systems (MAS) into a single model (formerly referred to as an "Agent Foundation Model"). The method involves generating multi-agent trajectories (Planner, Thinker, Reflector, etc.), filtering them, and using them for Supervised Fine-Tuning (SFT) and Reinforcement Learning (RL). The authors demonstrate that this approach significantly reduces inference costs compared to full MAS while achieving state-of-the-art performance on benchmarks like GAIA, BrowseComp, and MATH500. The rebuttal phase was highly productive, with the authors conducting significant additional experiments involving newer backbone models (Qwen 3), prompt-only baselines, and generalizations to new tools.

**Reviewer Concerns:**

Adressed:
- Source of Performance Gains (Reviewer qAjm, fCmh)
- Novelty and Comparison to Search-R1 (Reviewer AH31, fCmh)
- Timeliness/Backbone Model (Reviewer fCmh)
- Flexibility (Reviewer AH31)

Outstanding:
- Incremental Technical Novelty
- Still depend on LLM-as-a-Judge

**Reviewer Scores:**

Reviewer fCmh might raise the score to 5/6, who is also the most active reviewer during the rebuttal.

---

### Decision · Program_Chairs · 2026-01-26

Reject